# IGF1 and Insulin Receptor Single Nucleotide Variants Associated with Response in HER2-Negative Breast Cancer Patients Treated with Neoadjuvant Chemotherapy with or without a Fasting Mimicking Diet (BOOG 2013-04 DIRECT Trial)

**DOI:** 10.3390/cancers15245872

**Published:** 2023-12-17

**Authors:** Nadia de Gruil, Stefan Böhringer, Stefanie de Groot, Hanno Pijl, Judith R. Kroep, Jesse J. Swen

**Affiliations:** 1Department of Medical Oncology, Leiden University Medical Center, 2333 ZA Leiden, The Netherlands; s.de_groot.int@lumc.nl; 2Department of Medical Statistics and Bioinformatics, Leiden University Medical Center, 2333 ZA Leiden, The Netherlands; s.boehringer@lumc.nl; 3Department of Endocrinology, Leiden University Medical Center, 2333 ZA Leiden, The Netherlands; h.pijl@lumc.nl; 4Department of Clinical Pharmacy and Toxicology, Leiden University Medical Center, 2333 ZA Leiden, The Netherlands; j.j.swen@lumc.nl

**Keywords:** breast cancer, *IGF1R*, insulin pathway, biomarkers, neoadjuvant chemotherapy, fasting mimicking diet

## Abstract

**Simple Summary:**

Insulin and insulin-like growth factor 1 (IGF1) are metabolic hormones, which are often upregulated to stimulate proliferation in breast cancer. A fasting mimicking diet (FMD) targets insulin signaling pathway downregulation to hamper tumor growth. Genes encoding for the insulin receptors on the cell’s surface contain genetic variation between patients, which can affect insulin receptor function and cellular response. Therefore, a group of 113 patients with HER2-negative breast cancer receiving neoadjuvant chemotherapy with or without a fasting mimicking diet were investigated. We found that two IGF1 receptor variants were associated with worse pathological response compared to the reference alleles, out of the 17 interrogated common variants. Additionally, two IGF1 receptor variants could interact negatively within the FMD group regarding radiological response. These results emphasize that genetic variation harbors predictive clinical relevance to optimize and personalize cancer therapy.

**Abstract:**

Aim: We aimed to investigate associations between *IGF1R* and *INSR* single nucleotide variants (SNVs) and clinical response in patients with breast cancer treated with neoadjuvant chemotherapy with or without a fasting mimicking diet (FMD) from the DIRECT trial (NCT02126449), since insulin-like growth factor 1 (IGF1) and the insulin pathway are heavily involved in tumor growth and progression. Methods: Germline DNA from 113 patients was tested for 17 systematically selected candidate SNVs in *IGF1R* and *INSR* with pathological and radiological response. Results: *IGF1R* variants A > G (rs3743259) and G > A (rs3743258) are associated with worse pathological response compared to reference alleles *p* = 0.002, OR = 0.42 (95%CI: 0.24; 0.73); *p* = 0.0016; OR = 0.40 (95%CI: 0.23; 0.70). *INSR* T > C (rs1051690) may be associated with worse radiological response *p* = 0.02, OR = 2.92 (95%CI: 1.16; 7.36), although not significant after Bonferroni correction. Exploratory interaction analysis suggests that *IGF1R* SNVs rs2684787 and rs2654980 interact negatively with the FMD group regarding radiological response *p* = 0.036, OR = 5.13 (95%CI: 1.12; 23.63); *p* = 0.024, OR = 5.71 (95%CI: 1.26; 25.85). Conclusions: The *IGF1R* variants rs3743259 and rs3743258 are negatively associated with pathological response in this cohort, suggesting potential relevance as a predictive biomarker. Further research is needed to validate these findings and elucidate the underlying mechanisms and interaction with FMD.

## 1. Introduction

The insulin-like growth factor 1 (IGF1) and insulin pathway are both involved in tumor proliferation and progression [1,2]. Elevated IGF1 levels are specifically associated with increased risk of breast cancer [3], and high IGF1 levels are associated with increased breast cancer mortality, with a hazard ratio of 3.1. Increased IGF1 receptor (IGF1R) expression is found in 50% of breast cancers. Therefore, it is hypothesized that genetic variation affecting the IGF1/insulin axis may also influence cancer risk, progression and therapy response [4]. 

De Groot et al. previously showed that the *IGF1R* SNV G > T rs2016347 is associated with pathological response after neoadjuvant chemotherapy in patients with human epidermal growth factor receptor 2 (HER2)-negative breast cancer, emphasizing that genetic variation could impact treatment response in these patients [5]. The insulin receptor (*INSR*) gene has been studied less extensively in cancer, even though its protein can bind the same ligands as the IGF1R.

Moreover, *IGF1R* and *INSR* SNVs may influence the effects of short-term fasting or a fasting mimicking diet (FMD), since the FMD is suggested to operate at least partially through the IGF1 and insulin pathways [6]. Fasting has repeatedly been shown to have anti-cancer effects in preclinical research by sensitizing tumor cells for chemotherapy [7,8,9]. The underlying mechanism of FMD on the anti-cancer effect is that a decrease in the blood concentration of glucose, insulin and IGF1 causes IGF1R- protein kinase B(Akt)-mammalian target of rapamycin (mTOR) pathway downregulation [10,11], which leads to gene expression profile alterations that ultimately promote autophagy and cell death in cancer cells [12]. Subsequently, our phase 2 DIRECT study (NCT02126449) suggested a positive effect of the FMD compared to regular diet in addition to neoadjuvant chemotherapy on pathological and radiological response, in patients with early-stage HER2-negative breast cancer [13].

Therefore, we hypothesize that genetic variation affecting the IGF1/insulin axis may influence chemotherapy response and interact with FMD therapy, such as reported in the DIRECT study (NCT02126449). Here, we investigated *IGF1R* and *INSR* SNVs and, subsequently, IGF1R expression for association with pathological and radiological response.

## 2. Materials and Methods

The 131 patients who participated from February 2014 to January 2018 in the phase II randomized DIRECT trial (NCT02126449) were randomized to receive standard neoadjuvant chemotherapy with or without FMD [13]. Patient characteristics are shown in Table 1, and the study inclusion and exclusion criteria were described previously [13]. Further, 2 patients were excluded from analysis due to informed consent withdrawal and metastasis at inclusion. 

Pathological response was evaluated by the Miller–Payne (MP) score on a 1 to 5 scale [14]. Radiological response was assessed after chemotherapy and scored according to the RECIST1.1 criteria [15]. MP score was also grouped, with responders defined as score 4–5, less than 10% tumor cells, and the non-responders as score 1–3, as shown in Table 1. Radiological response is grouped as responders comprising complete response (CR) and partial response (PR) and the non-responders of stable disease (SD) and progressive disease (PD). The response data are primarily analyzed as intention-to-treat (ITT) and secondary in per protocol (PP) analysis with FMD-compliant versus control group, since 22 (33.8%) out of 65 FMD patients were able to comply with at least half of the planned FMD cycles. All patients provided written informed consent at the start of the study participation. The study was conducted in accordance with the Declaration of Helsinki (2008) and approved by the Medical Ethics Committee of the Leiden University Medical Center in agreement with the Dutch law for medical research involving humans. 

The 1000 Genomes database [16,17], version GRCh37p13, provided all SNVs for *IGF1R* (*n* = 1364) and *INSR* (*n* = 1244) genes. Selection criteria for SNV selection included (1) localization in exon positions, (2) minor allele frequency ≥0.2 in the sub-population with Northern and Western European ancestry (CEU) and (3) non-duplicates. These selection criteria resulted in a total of 24 SNVs, 15 for *IGF1R* and 9 for *INSR*. Due to technical limitations in primer design of the custom Open Array chip, 6 SNVs had to be replaced with proxy SNVs. Haploview software (version 4.1) identified 4 SNVs in high linkage disequilibrium *r*^2^ > 0.9, namely *IGF1R* rs1815009 for rs66745311, rs2684788 for rs3051367, rs2654980 for rs9282714 and *INSR* rs2252673 for rs2352955 (Appendix A). INSR rs34045095 and rs2352954 had to be excluded due to lack of proxy SNV and internal quality control. Ultimately, 17 candidate SNVs were selected, 11 in *IGF1R* and 6 in *INSR* (Appendix A).

DNA was isolated from baseline blood samples (*n* = 113) collected in Ethylene diamine tetra acetic acid tubes stored at from −80 °C. Isolated DNA samples were stored at −20 °C until genotyping for the 17 candidate *IGF1R* and *INSR* SNVs. Genotyping was performed using a PCR-based fixed-format OpenArray™ Panel (Thermo Fisher Scientific, Waltham, MA, USA) to detect SNVs using specific probes for the genes *IGF1R* and *INSR*. Reactions were run on the QuantStudio™ 12 K Flex OpenArray Genotyping system (Thermo Fisher Scientific, Waltham, MA, USA) and analyzed with the TaqMan Genotyper Software^®^ (version 1.3). The predefined minimum call rate was >85%. 

Formalin-fixed paraffin-embedded blocks of the diagnostic biopsies and resection material were sectioned (4 µm) and immunohistochemically stained for membranous IGF-1R expression, as described elsewhere more extensively [5]. For positive controls, placenta tissue with previously confirmed IGF1R expression was used, while negative control sections underwent the same IHC procedure without the primary antibody. Scoring the membranous IGF1R expression was performed by two assessors (SdG, NdG) simultaneously and, if necessary, sections were checked by a pathologist to reach consensus. The scoring method, as described elsewhere in more detail [5], was, in short, carried out on a scale from 0 to 3+. A score 0 was given if <10% of the tumor cells were incompletely stained, 1 if >10% of tumor cells showed incomplete staining, 2 if weak to moderate staining in >10% of the tumor cells was observed and 3+ if strong complete staining was observed in >10% of tumor cells. A score of 0 and 1+ was considered negative and 2+ and 3+ as positive [5]. Statistical analysis was performed in Statistical Package for Social Sciences (IBM SPSS, version 24.0 and 25.0, Armonk, NY, USA: IBM Corp). Genotype distribution in 1000 Genomes and the DIRECT cohort were compared and tested for deviation from Hardy–Weinberg equilibrium (HWE) using a goodness-of-fit test with *p*-value of <0.05 as significance threshold. Ordinal and binary logistic regression using univariate and multivariate models was used for pathological and radiological response and IGF1R expression. The proportional odds assumption was not violated for the significant SNVs, as assessed by comparing regression coefficients to a separate multinomial regression analysis (Appendix A). For the primary analysis, the ITT population was used. First, model selection was performed with univariate regressions performed on potential confounders and influential variables. Variables with *p*-values < 0.1 were carried forward into the primary multivariate analysis. In the genetic association models, genotypes were used with additive coding. Coding of genotypes was performed according to the variant allele, i.e., the genotype represents how often the variant allele is present, and associations are interpreted in terms of the variant allele. In secondary analyses, the PP population was considered. Further analyses were conducted in an explorative way to investigate possible interactions between SNV and treatment group. For the primary analysis, Bonferroni correction was applied to account for multiple testing of 17 SNVs, with a significance threshold of 0.05/17 = 0.003.

## 3. Results

### 3.1. IGF1R and INSR SNV Distribution

Baseline blood samples from 113 out of 131 patients were available for analysis. The SNV distribution among the study cohort shown in Table 2 is comparable to the frequencies observed in the publicly available databases of PubMed and 1000 Genomes GRCh37p13 (Appendix A). Furthermore, all the SNVs followed HWE at the 0.05 threshold. The predetermined minimum call rate of >85% was achieved with a minimum of 92% (Appendix A).

### 3.2. IGF1 Receptor SNVs Are Associated with Worsened Pathological Response and INSR SNV Is Potentially Associated with Worse Radiological Response

In the model selection step, tumor and lymph node status, age, randomization and hormone receptor (HR) status were selected as covariates for the primary analysis (Appendix A). Tumor type and HR status are biologically similar factors; therefore, only HR status was entered into the final ordinal regression model to optimize the noise-to-signal ratio. 

*IGF1R* rs3743259 and rs3743258 SNVs were associated with worse pathological response compared to the reference genotype in the ITT analysis with an additive model, *p* = 0.002, OR = 0.42 (95%CI: 0.24; 0.73); *p* = 0.0016; OR = 0.40 (95%CI: 0.23; 0.70), respectively, as shown in Table 3.

Multivariate analysis in a PP fashion yielded associations with the same SNVs and similar effect sizes, though these were not statistically significant after Bonferroni correction for multiple testing (See Table 4).

Furthermore, *INSR* rs1051690 presence suggested association with worse radiological response compared to patients with the reference genotype (Table 3: OR = 2.92 (95%CI: 1.16; 7.36); *p* = 0.02), although this association was not significant after Bonferroni correction. The logistic regression model with responders vs. non-responders for radiological and pathological response showed no significant correlation after correction for multiple testing for ITT, which suggests that the ordinal regression analysis best retains information on the association (Appendix A).

### 3.3. IGF1R SNVs and FMD Interaction

Further ordinal regression analyses were conducted to analyze potential interaction between SNVs and FMD. This analysis suggested that the presence of *IGF1R* SNVs rs2684787 and rs2654980 might interact with the ITT FMD group differently compared to the control group, affecting radiological response negatively, *p* = 0.036, OR = 5.13 (95%CI: 1.12–23.63); *p* = 0.024, OR = 5.71 (95%CI: 1.26–25.85), respectively (Appendix A), but not in PP. For pathological response, there was no indication of interaction in the ITT or PP.

Secondary models included responders vs. non-responders, which revealed that there were no indications for interaction between SNVs and treatment group affecting radiological response and pathological responders vs. non-responders after correction for multiple comparison (Appendix A). 

### 3.4. IGF1R Expression Score Is Not Associated with Clinical Response

We found that 58 of the total 104 biopsies (55.8%) were IGF1R-positive at baseline; 28 positive IGF1R biopsies at baseline became negative (48.3% of positive biopsies; 26.9% of total), while 30 remained IGF1R positive (51.7% of positive biopsies; 28.8% of total) at resection, as seen in Figure 1. Further, 44 of 46 (42.3% of total) biopsies remained IGF1R negative at resection (95.7% negative biopsies; 42.3% of total), while 2 became positive (4.3% of negative biopsies; 1.9% of total) (see Figure 1). Multivariate regression analysis uncovered no association between IGF1R biopsy or resection score and pathological or radiological response in both the ITT and PP multivariate model.

## 4. Discussion

This study shows that the presence of *IGF1R* SNVs rs3743259 and rs3743258 is associated with worse pathological response compared to the reference genotype in this cohort of patients with breast cancer treated with neoadjuvant chemotherapy (see Table 3). Additionally, our results suggest that *INSR* rs1051690 may be associated with worse radiological response compared to patients with the reference genotype in this study, although this association was not significant after Bonferroni correction. These findings indicate that patients with rs3743259 and rs3743258 may respond less to chemotherapy, which could imply treatment consequences, for example, to consider an alternative systemic therapy regimen. The association between rs3743259 and rs3743258 was not significant in the PP analysis, most likely due to the smaller sample size of 72 compared to 112 in the ITT analysis. Furthermore, there might be an interaction between FMD and the presence of SNVs, as shown by the exploratory analysis, and *IGF1R* SNVs rs2684787 and rs2654980 might interact with the FMD group differently compared to the control group regarding radiological response (Appendix A). 

To assess whether the observed associations could be attributed to other genes as indicated by SNVs exhibiting high linkage disequilibrium (LD), online tools LDlink and Haploreg (version 4.1) were used [18,19]. Firstly, *IGF1R* rs3743259 shows high LD with six other SNVs (*r*^2^ > 0.8), including *IGF1R* rs3743258 (*r*^2^ = 0.9). All six SNVs have not been associated with disease or pathologic processes according to PubMed. Furthermore, LDlink reports that *IGF1R* rs3743258 is in LD with five SNVs with an *r*^2^ > 0.8, whereas HaploReg only reports rs3743259. Nevertheless, all associated SNVs hail from the *IGF1R* gene, and it is, therefore, unlikely that the observed effect is originating from another gene or SNV. 

Biong et al. found that rs3743259 was associated with increased mammographic density [20], which is a known risk factor for breast cancer development [21]. Together with our findings on rs3743259, this study supports the hypothesis that there could be a biological mechanism to drive breast cancer. Nevertheless, the structural and functional consequences of these SNVs on IGF1 receptor signaling have yet to be investigated.

The limitations of this study are the relatively small sample size for a genetic association study. In order to decrease the risk of reporting false-positive results, we applied a strict Bonferroni correction. The statistical limitation of this study is illustrated by the exploratory generalized linear models used to investigate the interaction between SNVs and FMD versus SNVs and control group. These results should be interpreted carefully, since analyzing the cohort in smaller groups or more complex models decreases statistical power. Therefore, the interaction between the *IGF1R*/*INSR* SNVs and FMD is important to investigate in future studies. Furthermore, radiological response data were missing for 21/129 patients (16.3%), although separate regression analysis confirmed that these missing data did not arise due to other variables. 

Lastly, future research could focus on the structural effect of these *IGF1R* SNVs on the protein in silico, for example. Subsequently, the functional consequences, in terms of receptor affinity for ligands, should be investigated as well. In parallel, the findings of this study need validation in another, preferably larger, breast cancer cohort or a genetic databank cohort. Moreover, future studies, such as the ongoing phase III DIRECT2 (NCT05503108) study, are necessary to validate the currently reported associations. In conclusion, more translational research is needed to reveal the underlying mechanisms between the genetic variation in *IGF1R* and *INSR* in the context of fasting and clinical response in cancer.

## 5. Conclusions

This study identified *IGF1R* SNVs rs3743259 and rs3743258 as potential predictive markers for worse pathological response on neoadjuvant chemotherapy in patients with HER2-negative breast cancer. Validation and further research are essential before any clinical recommendations can be made.

## Figures and Tables

**Figure 1 cancers-15-05872-f001:**
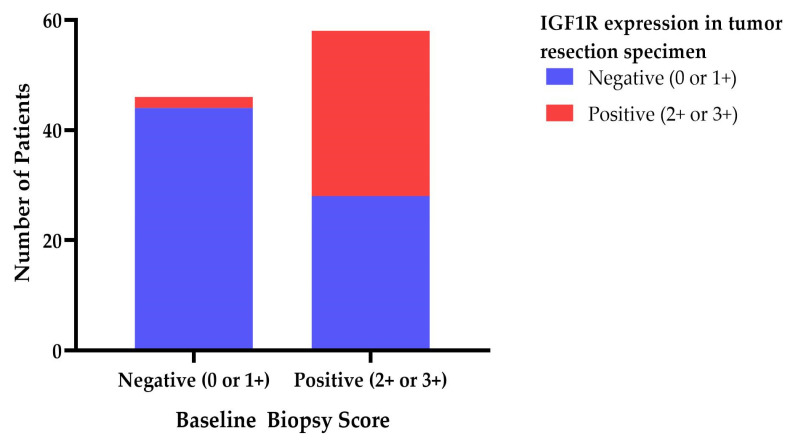
IGF1R expression in baseline tumor biopsy and tumor resection specimen.

**Table 1 cancers-15-05872-t001:** Patient characteristics.

Characteristics	Label	*n* Cases (Median)	%	Missing Cases	Total Cases
Median Age	(range) years	50 (27–71)		0	129
Median BMI	(range) kg/m^2^	25.8 (19.7–41.2)		0	129
Randomization	chemo + FMD	65	50.4%	0	129
	chemo	64	49.6%		
Per-protocol *	chemo + FMD compliant	22	17.1%	0	124
	chemo + FMD non-compliant	43	34.7%		
	chemo	59	47.6%		
HR status	ER−/Progesterone−	21	16.3%	1 (0.8%)	128
	ER+/Progesterone−	18	14.0%		
	ER+/Progesterone+	89	69.8%		
Tumor Type	Other	5	3.5%	0	129
	Lobular	22	20.2%		
	Ductal/Carcinoma	102	76.3%		
Tumor status **	cT1	11	8.4%	0	129
	cT2	83	64.9%		
	cT3	32	24.4%		
	cT4	3	2.3%		
Lymph node status **	cN0	63	48.1%	0	129
	cN1	55	42.0%		
	cN2	11	8.4%		
	cN3	2	1.5%		
Miller&Payne score	grade 1 no reduction	35	27.1%	1 (0.8%)	128
	grade 2 < 30% tumor reduction	26	20.9%		
	grade 3 30–90% tumor reduction	33	25.6%		
	grade 4 > 90% tumor reduction	20	15.5%		
	grade 5 no tumor	14	10.9%		
Miller&Payne pooled	grade 1–3 non-responders	94	73.3%	2 (1.5%)	128
	grade 4–5 responders	34	26.4%		
Radiological response	CR	16	14.8%	23 (17.6%)	108
	PR	69	63.9%		
	SD	22	20.4%		
	PD	1	0.9%		
Radiological response pooled	CR or PR responders	85	78.7%	23 (17.6%)	108
	SD or PD non-responders	23	21.3%		

* Per protocol groups consisted of chemo + FMD compliant (≥half of the planned FMD cycles), chemo + FMD non-compliant (≤half of the planned FMD cycles) and the control group chemo. ** Tumor and lymph node status according to TNM classification. BMI body mass index (kg/m^2^). HR-status hormone receptor status. ER estrogen receptor. Progesterone progesterone receptor. CR complete response. PR partial response. SD stable disease. PD progression of disease.

**Table 2 cancers-15-05872-t002:** SNV distribution.

SNV	Reference Genotype	Heterozygous Genotype	Homozygous Genotype	MAF Ref Allele	N	HWE	*p*-Value *
IGF1R rs2016347	32	56	25	53%	113	0.003	0.957
IGF1R rs2229765	35	60	17	58%	112	1.116	0.291
IGF1R rs1815009	5	49	59	26%	113	1.735	0.188
INSR rs1051651	78	28	6	82%	112	2.447	0.118
INSR rs3745551	12	50	50	33%	112	0.009	0.924
IGF1R rs3743259	58	41	12	71%	111	1.297	0.255
IGF1R rs2684787	60	42	10	72%	112	0.449	0.503
IGF1R rs2654981	25	55	33	46%	113	0.053	0.818
IGF1R rs2654980	61	42	10	73%	113	0.499	0.480
IGF1R rs2684788	27	55	31	48%	113	0.072	0.788
IGF1R rs3743249	59	47	7	73%	113	0.346	0.556
IGF1R rs45484096	51	46	16	65%	113	1.118	0.290
INSR rs3833238	77	30	6	81%	113	1.700	0.192
INSR rs1051690	2	29	82	15%	113	0.095	0.757
INSR rs1799817	85	24	4	86%	113	1.802	0.179
INSR rs2252673	3	23	82	13%	108	0.760	0.383
IGF1R rs3743258	57	35	12	72%	104	3.072	0.080

SNV single nucleotide variant, *INSR* insulin receptor gene, *IGF1R* insulin like growth factor 1 receptor gene, MAF mean allele frequency, HWE Hardy–Weinberg equation. * if <0.05—not consistent with HWE. Call rate minimum is 92%.

**Table 3 cancers-15-05872-t003:** Primary multivariate ordinal regression model intention to treat.

		Miller & Payne						Radiological Response				
SNVs	N	OR	95%CI Lower-Upper			*p*-Value	N	OR	95%CI Lower-Upper			*p*-Value
IGF1R rs2016347	112	1.52	0.94	-	2.46	0.09	92	0.89	0.48	-	1.67	0.72
IGF1R rs2229765	111	1.03	0.61	-	1.73	0.91	91	1.14	0.58	-	2.25	0.70
IGF1R rs1815009	112	0.83	0.46	-	1.50	0.53	92	0.92	0.43	-	1.98	0.84
INSR rs1051651	111	1.46	0.82	-	2.62	0.20	91	1.69	0.80	-	3.59	0.17
INSR rs3745551	111	0.99	0.60	-	1.66	0.98	91	1.88	0.93	-	3.77	0.08
IGF1R rs3743259	110	0.42	0.24	-	0.73	**0.002**	91	1.10	0.56	-	2.16	0.79
IGF1R rs2684787	111	1.23	0.73	-	2.07	0.44	91	0.92	0.47	-	1.80	0.80
IGF1R rs2654981	112	1.23	0.77	-	1.98	0.39	92	0.98	0.54	-	1.78	0.96
IGF1R rs2654980	112	1.25	0.75	-	2.11	0.39	92	0.88	0.45	-	1.72	0.71
IGF1R rs2684788	112	1.54	0.96	-	2.49	0.08	92	1.00	0.54	-	1.83	0.99
IGF1R rs3743249	112	1.39	0.79	-	2.45	0.25	92	0.95	0.47	-	1.94	0.89
IGF1R rs45484096	112	1.03	0.64	-	1.65	0.91	92	1.04	0.56	-	1.94	0.89
INSR rs3833238	112	1.32	0.75	-	2.35	0.34	92	1.75	0.83	-	3.67	0.14
INSR rs1051690	112	0.53	0.27	-	1.06	0.07	92	2.92	1.16	-	7.36	**0.02**
INSR rs1799817	112	0.78	0.40	-	1.49	0.45	92	1.89	0.74	-	4.81	0.18
INSR rs2252673	107	0.87	0.42	-	1.78	0.70	88	2.37	0.89	-	6.34	0.09
IGF1R rs3743258	103	0.40	0.23	-	0.70	**0.002**	86	1.12	0.57	-	2.20	0.75

Cut off value Bonferroni correction for multiple testing is 0.05/17 = 0.0029. OR odds ratio, CI confidence interval, SNV single nucleotide variant, *INSR* insulin receptor, *IGF1R* insulin like growth factor 1 receptor. Bold: statistically significant.

**Table 4 cancers-15-05872-t004:** Primary multivariate ordinal regression model per protocol.

SNVs	Miller & Payne						Radiological Response					
	N	OR	95%CI			*p*-Value	N	OR	95%CI			*p*-Value
IGF1R rs2016347	72	1.40	0.74	-	2.63	0.30	61	0.76	0.34	-	1.74	0.52
IGF1R rs2229765	72	1.21	0.63	-	2.33	0.57	61	0.93	0.41	-	2.13	0.87
IGF1R rs1815009	72	1.07	0.52	-	2.19	0.86	61	0.56	0.22	-	1.41	0.22
INSR rs1051651	72	1.11	0.56	-	2.21	0.77	61	1.67	0.71	-	3.94	0.24
INSR rs3745551	72	1.49	0.79	-	2.80	0.22	61	1.72	0.77	-	3.87	0.19
IGF1R rs3743259	72	0.49	0.25	-	0.94	0.03	61	1.24	0.56	-	2.72	0.59
IGF1R rs2684787	71	1.29	0.66	-	2.52	0.46	60	0.45	0.18	-	1.12	0.09
IGF1R rs2654981	72	0.98	0.51	-	1.88	0.94	61	0.91	0.42	-	2.00	0.82
IGF1R rs2654980	72	1.36	0.70	-	2.64	0.37	61	0.43	0.17	-	1.05	0.06
IGF1R rs2684788	72	1.33	0.71	-	2.50	0.37	61	0.85	0.38	-	1.87	0.68
IGF1R rs3743249	72	1.10	0.54	-	2.21	0.80	61	1.37	0.57	-	3.30	0.49
IGF1R rs45484096	72	1.07	0.59	-	1.94	0.83	61	0.68	0.31	-	1.49	0.33
INSR rs3833238	72	1.01	0.51	-	2.02	0.97	61	1.58	0.67	-	3.73	0.29
INSR rs1051690	72	0.41	0.16	-	1.04	0.061	61	3.41	1.09	-	10.63	0.035
INSR rs1799817	72	0.51	0.21	-	1.28	0.15	61	1.55	0.46	-	5.31	0.48
INSR rs2252673	69	1.30	0.56	-	3.04	0.54	59	1.68	0.56	-	5.02	0.35
IGF1R rs3743258	68	0.46	0.24	-	0.91	0.03	59	1.23	0.56	-	2.70	0.61

Cut off value Bonferroni correction for multiple testing is 0.05/17 = 0.0029. OR odds ratio, CI confidence interval, SNV single nucleotide variant, *INSR* insulin receptor, *IGF1R* insulin-like growth factor 1 receptor.

## Data Availability

The data that support the findings of this study are available from the corresponding author, J.R.K., upon reasonable request.

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
