# Peer review of "IGF1 and Insulin Receptor Single Nucleotide Variants Associated with Response in HER2-Negative Breast Cancer Patients Treated with Neoadjuvant Chemotherapy with or without a Fasting Mimicking Diet (BOOG 2013-04 DIRECT Trial)"

_cancers, 2023, doi:10.3390/cancers15245872_

Round 1
Reviewer 1 Report
Comments and Suggestions for Authors
Nadia de Gruil and colleagues described in their report the results of study on association between some insulin-like growth factor 1 (IGF1) and insulin receptor (INSR) genetic variants and clinical response to neoadjuvant chemotherapy of patients with HER-2 negative breast cancer with or without of fasting mimicking diet. In general, the study is well planned and executed. The methods are appropriate and the manuscript is properly structured and written. This is an interesting study, however have some drawbacks. In my opinion the major limitation of the study is the small number of investigated patients which do not allowed authors to achieve stronger associations. I fully agree with the authors that the findings of this study need confirmation in another study performed on larger sample.
Author Response
Thank you for your time and kind comments. Because of the retrospective setting of the already finished trial, we were unable to include more patients. As you state, we agree that the study should be validated in a larger similar study sample
Reviewer 2 Report
Comments and Suggestions for Authors
Thank you very much for this paper. I find it interesting, clear, and direct to the topic.
I hope to see the future findings regarding the structural effects and the potential therapeutic strategies.
Author Response
Thank you for your time and the kind comments on the paper.
Reviewer 3 Report
Comments and Suggestions for Authors
In the article of de Gruil and colleagues, the authors investigate the association between single nucleotide variants on IGF1R and IR genes and clinical response in HER-2-negative breast cancer patients.
The study is interesting, well designed, and it provides novel biomarkers of response in breast cancer.
I have some minor comments:
-To this reviewer is not clear how was IGF1R expression measured. Methods and results sections should include this part.
- Was IR expression measured as well?
-Minor typos are as follows: a comma after 2 (page 1, lines 22), missing “o” in progression (page 1, line 28), missing “F” in IG1R (page 5, line 158)
Author Response
Thank you for your time and the kind comments to improve the paper. Here we provide a point-by-point response:
1. To this reviewer is not clear how was IGF1R expression measured. Methods and results sections should include this part.
Response: We agree that the method for IGF1R expression analysis needs to be explained. A paragraph was added to address the sample setup while also referring to another paper from our group (de Groot S., Breast Cancer Res. 2016 Jan 6) that contains a more extensive lab protocol of the IGF1R staining. Secondly the scoring method is summarized, as well as referred to a more detailed explanation in the same paper (de Groot S., Breast Cancer Res. 2016 Jan 6).
2. Was IR expression measured as well?
Response: IR expression was not stained or measured.
3. Minor typos are as follows: a comma after 2 (page 1, lines 22), missing “o” in progression (page 1, line 28), missing “F” in IG1R (page 5, line 158)
Response: Thank you, we have modified the manuscript accordingly.